# ProSST: Protein Language Modeling with Quantized Structure and Disentangled Attention

**Mingchen Li**[2,3,5*]   **Yang Tan**[2,3,5,*]   **Xinzhu Ma**[2,4]   **Bozitao Zhong**[1,4]   **Huiqun Yu**[3]
**Ziyi Zhou**[1]   **Wanli Ouyang**[2,4]   **Bingxin Zhou**[1]   **Pan Tan**[1,2]   **Liang Hong**[1,2,5]

[1] Shanghai Jiao Tong University, China
{zy-zhou,bingxin.zhou,hongl3liang}@sjtu.edu.cn, tpan1039@gmail.com,
[2] Shanghai Artificial Intelligence Laboratory, China
{ouyang-wanli,maxinzhu}@pjlab.org.cn
[3] East China University of Science and Technology, China
{lmc,tyang}@mail.ecust.edu.cn, yhq@ecust.edu.cn
[4] The Chinese University of Hong Kong, China
zbztzhz@gmail.com;
[5] Chongqing Artificial Intelligence Research Institute of Shanghai Jiao Tong University, China

## Abstract

Protein language models (PLMs) have shown remarkable capabilities in various protein function prediction tasks. However, while protein function is intricately tied to structure, most existing PLMs do not incorporate protein structure information. To address this issue, we introduce ProSST, a Transformer-based protein language model that seamlessly integrates both protein sequences and structures. ProSST incorporates a structure quantization module and a Transformer architecture with disentangled attention. The structure quantization module translates a 3D protein structure into a sequence of discrete tokens by first serializing the protein structure into residue-level local structures and then embeds them into dense vector space. These vectors are then quantized into discrete structure tokens by a pre-trained clustering model. These tokens serve as an effective protein structure representation. Furthermore, ProSST explicitly learns the relationship between protein residue token sequences and structure token sequences through the sequence-structure disentangled attention. We pre-train ProSST on millions of protein structures using a masked language model objective, enabling it to learn comprehensive contextual representations of proteins. To evaluate the proposed ProSST, we conduct extensive experiments on the zero-shot mutation effect prediction and several supervised downstream tasks, where ProSST achieves the state-of-the-art performance among all baselines. Our code and pre-trained models are publicly available [2].

## 1   Introduction

Predicting the functions of proteins is one of the most critical areas in life sciences [1]. In recent decades, protein sequence databases have experienced exponential growth [2], making it possible to learn the fundamental representations of protein sequences with large-scale models in a data-driven manner. Inspired by pre-trained language models in natural language processing [3, 4], many pre-trained Protein Language Models (PLMs) have emerged [5, 6, 7, 8, 9]. Benefiting from

---

*These authors contributed equally to this work, and this work was done during their internship at the Shanghai Artificial Intelligence Laboratory. Corresponding authors: Bingxin Zhou (bingxin.zhou@sjtu.edu.cn), Pan Tan (tpan1039@gmail.com) and Liang Hong (hongl3liang@sjtu.edu.cn)

[2]https://github.com/ai4protein/ProSST

remarkable protein representation capabilities, they have become fundamental tools for bioinformatics in protein-related tasks.

The function of a protein is determined by its structure [10]. However, most PLMs mainly focus on modeling protein sequences, neglecting the importance of structural information, and one significant reason for this phenomenon is the lack of structural data. Fortunately, some excellent works, such as AlphaFold [11] and RoseTTAFold [12], are proposed, which can accurately predict protein structures. These works significantly expand the protein structure dataset [13] to millions and enable the pre-training of large-scale structure-aware PLMs. After that, the major challenge is how to effectively integrate protein structure information into PLMs. Specifically, existing structure-aware PLMs [14, 15] first use Foldseek [16] to convert protein structures into discrete structure tokens and then integrate these structural data into the Transformer architecture. However, despite achieving promising performance on several tasks, this approach still faces two main issues. First, Foldseek encodes the structure of a residue within a protein by considering only the features of its previous and next residues. This representation is insufficient and may overlook subtle differences in the local structure of proteins, such as catalytic sites or binding pockets, which are crucial for protein function [17]. Second, the naive Transformer architecture lacks the ability to explicitly model the relationship between protein sequences and structure token sequences, making it challenging to effectively leverage structural cues.

In this paper, we develop ProSST (**Pro**tein **S**equence-**S**tructure **T**ransformer), a structure-aware pre-trained protein language model. Specifically, ProSST mainly consists of two modules: a structure quantization module and a Transformer with sequence-structure disentangled attention. The structure quantization module is based on a GVP (Geometric Vector Perceptron) [18] encoder, which can encode a residue structure along with its neighborhoods in its local structure and quantize the encoding vectors into discrete tokens. Compared to Foldseek, which only considers individual residues, this encoder can take into account more information from the micro-environment of residue. The sequence-structure disentangled attention module replaces the self-attention module in the Transformer model. This can make Transformer model explicitly model the relationship between protein sequence tokens and structure tokens, enabling it to capture more complex features of protein sequences and structures. To enable ProSST to learn the contextual representation comprehensively, we pre-train our model with the Masked Language Modeling (MLM) objective on a large dataset containing 18.8 million protein structures. To summarize, our main contributions are as follows:

- We propose a protein structure quantizer, which can convert a protein structure into a sequence of discrete tokens. These token sequences effectively represent the local structure information of residues within a protein.

- We propose a disentangled attention mechanism to explicitly learn the relationship between protein structure and residue, facilitating more efficient integration of structural token sequences and amino acid sequences.

To evaluate the proposed ProSST, we conduct extensive experiments on zero-shot mutation effect prediction and multiple supervised downstream tasks, where the proposed model achieves state-of-the-art results among all baselines. Besides, we also provide detailed ablations to demonstrate the effectiveness of each design in ProSST.

## 2 Related Work

### 2.1 Protein Representation Models

Based on the input modality, protein representation models can be divided into three categories: sequence-based models, structure-based models, and structure-sequence hybrid models.

**Sequence-based models.** Sequence-based models treat proteins as a sequence of residue tokens, using the Transformer model [19] for unsupervised pre-training on extensive datasets of sequence. According to the pre-training objective, current models can be further divided into BERT-based models [4], GPT-based models [3], and span-mask based models. Specifically, BERT-style models, including ESM-series models [5, 6, 7], ProteinBert[9], and TAPE [20], aim to recover the masked tokens in the training phase. The GPT-style models, such as Tranception [21], ProGen2 [22], and

ProtGPT2 [23], progressively generate the token sequences in an auto-regressive manner. Lastly, models that use span-mask as the training objective include Ankh [24], ProtT5 [8], and xTrimo [25].

**Structure-based models.** Protein structures play a dominant role in protein functionality. Therefore, models leveraging structure information generally get more accurate predictions. Recently, various techniques have been applied in learning protein structure representation, including CNN-based models [26] and GNN-based models [18, 27, 28, 29, 30], where the GNN-based ones have demonstrated significant versatility in integrating protein-specific features through node or edge attributes. Moreover, the recent advancements in protein folding models [7, 11, 31] enable the structure-based models access to extensive datasets of protein structures. This led to a growing interest in developing PLMs that leverage protein structure cues [14, 15, 32].

**Structure-sequence hybrid models.** Hybrid models, which incorporate both sequence and structure information of proteins, offer more effective representations of proteins. For example, the LM-GVP[33] model employs ProtBERT-BFD [9] embeddings as input features for the GVP [18] model, while ESM-GearNet [34] investigates various methods of integrating ESM-1b [5] representations with GearNet [32]. Similarly, the recent ProtSSN [35] model leverages ESM-2 [7] embeddings as input for the EGNN [36] model, resulting in notable advancements. Both ESM-IF1 [37] and MIF-ST [38] target inverse folding, utilizing the structure to predict corresponding protein residues, whereas ProstT5 [15] focuses on the transformation between residue sequences and their structure token sequences [16] as a pre-training objective. SaProt [14] constructs a structure-aware vocabulary using structure tokens generated by foldseek [16]. Both SaProt and ProstT5 extensively utilize large structure databases [13] for their pre-training datasets. ProSST is also a hybrid structure-sequence model. Compared to previous work, ProSST develops an advanced structure quantization method and a better attention formulation to leverage the structure cues.

## 2.2 Protein Structure Quantization

The most intuitive way to represent a protein structure is using continuous features, such as coordinates, dihedral angles and distance map. However, directly using these continuous features in the pre-training may lead to overfitting [14]. This issue arises from the mismatched representations of the structure between the training set (derived from model predictions) and the test set (measured by wet-lab experiments). As the bridge to eliminate this gap, structure quantization has been investigated by a few works. These methods can be divided into two groups based on the way to generate the discrete secondary structure, including the methods based on physical computing, such as DSSP [39], and the methods based on deep learning, such as Foldseek [16], which have been successfully applied to structure-aware PLMs [14, 15]. The structure quantization module of ProSST also relies on learning-based approaches but provides a more detailed residue structure representation than Foldseek.

## 3 Method

In this section, we introduce the architecture of ProSST. ProSST mainly contains two modules: structure quantization (Section 3.1) module and a-transformer-based model with sequence-structure disentangled attention. (Section 3.2).

### 3.1 Structure Quantization Module

The structure quantization module aims to transform a residue's local structure into a discrete token. Initially, the local structure is encoded into a dense vector using a pre-trained structure encoder. Subsequently, a pre-trained k-means clustering model assigns a category label to the local structure based on the encoded vector. Finally, the category label is assigned to the residue as the structure token. The pipeline of structure quantization is shown in Figure 1.

**Structure representation**. We categorize protein structures into two distinct levels: *protein structure* and *local structure*. Protein structure denotes the complete architecture of a protein, including all its residues. The local structure focuses on specific individual residues. It describes the local environment of a residue by centering on a specific residue and including it along with the nearest 40 residues surrounding it in three-dimensional space [18]. Compared to protein structure, local structures are in finer granularity, which allows for a more accurate description of the structure of

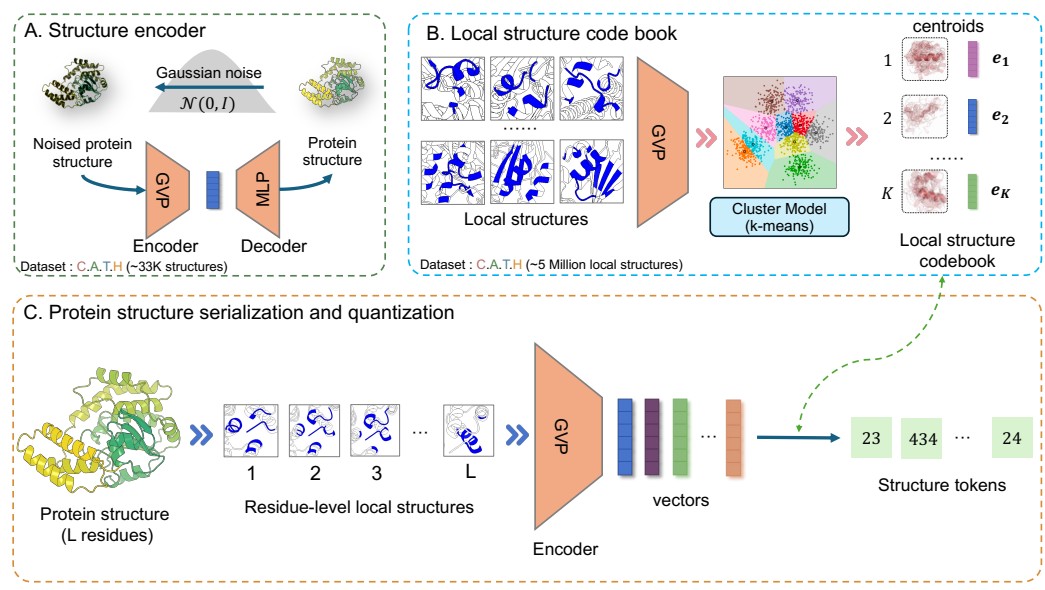

Figure 1: **The pipeline of structure quantization.** (A) Training of the structure encoder. (B) Local structure clustering and labeling. (C) Converting a protein structure to structure token sequence.

residue. Therefore, a protein containing $L$ residues has one protein structure and $L$ local structures. Despite the different levels of structure, we can use graphs to represent it. Formally, we represent a structure using graph $\boldsymbol{G} = (\boldsymbol{V}, \boldsymbol{E})$, where $\boldsymbol{V}$ and $\boldsymbol{E}$ denote the residue-level nodes and edges, respectively. For any given node $\boldsymbol{v} \in \boldsymbol{V}$, it contains only the structure information of the residue, without any residue type information of the residue itself. This ensures that the structure encoder is solely focused on the structure cues. The edge set $\boldsymbol{E} = \{\boldsymbol{e}_{ij}\}$ includes all $i, j$ for which $\boldsymbol{v}_j$ is one of the top-$40$ nearest neighbors of $v_i$, determined by the distance between their $\boldsymbol{C}\alpha$ atoms.

**Structure encoder**. Based on the above-mentioned definition of structure, we use geometric vector perceptrons (GVP) [18] as the (local) structure encoder. In particular, the GVP can be represented as a structure feature extraction function $\pi_\theta(\boldsymbol{G}) \in \mathbb{R}^{l \times d}$, where $l$ is the number of nodes, $d$ is the embedding dimension, and $\theta$ is trainable parameters. We integrate GVP with a decoder that includes a position-wise multi-layer perceptron (MLP) to form an auto-encoder model. The entire model is trained using a de-noising pre-training objective. In this process, we perturb $\boldsymbol{C}\alpha$ coordinates with 3D Gaussian noise (Figure 1A) and use Brownian motion on the manifold of rotation matrices, according to RF-Diffusion [40]. The model is then tasked with recovering the structure to its original, noise-free state. After being trained on the C.A.T.H dataset [41] (see Appendix A.2), we exclude the decoder and utilize solely the mean pooled output of the encoder as the final representation of structures. Although the structure encoder is trained on protein structures, it can effectively encode local structures. Therefore, for a graph $\boldsymbol{G}$ of a protein structure, the encoding is: $\boldsymbol{r} = \frac{1}{l} \sum_{i=1}^{l} \pi_\theta(\boldsymbol{g}_i)$, where $\boldsymbol{g}_i$ represents the graph of the local structure associated with the $i$-th residue in the graph $\boldsymbol{G}$, and $\pi_\theta(\boldsymbol{g_i}) \in \mathbb{R}^d$ is the output of the encoder for the $i$-th node. Here, $\boldsymbol{r} \in \mathbb{R}^d$ is the mean pooled output of the encoder and the vectorized representation of the local structure.

**Local structure codebook**. The structure code book quantizes dense vectors representing protein structure into discrete tokens (Figure 1B). To build this, we employ a structure encoder to embed the local structures of all residues from the C.A.T.H dataset (See in Appendix A.2) into a continuous latent space. Then we apply the $k$-means algorithm to identify $K$ centroids within this latent space, denoted as $\{\boldsymbol{e}_i\}_{i=1}^{K}$. These centroids constitute the structure codebook, as shown in Figure 1B. For any local-structure embedding, it is quantized by the nearest vector $\boldsymbol{e}_j$ within the codebook and $j$ serving as the structure token. In this paper, the clustering number $K$ is also referred to as the structure vocabulary size.

**Protein serialization and quantization.** In general, for a residue at position $i$ in a protein sequence, we first build a graph $\boldsymbol{g}_i$ only based on its local structure, and then use the structure encoder to embed

it into a continuous vector $\boldsymbol{r}_i$. Then we use the codebook to assign a structure token $s_i \in \{1, 2, ..., K\}$ to this vector as the structure token of the residue. Overall, the entire protein structure can be serialized and quantized into a sequence of structure tokens (Figure 1C).

## 3.2  Sequence-Structure Disentangled Attention

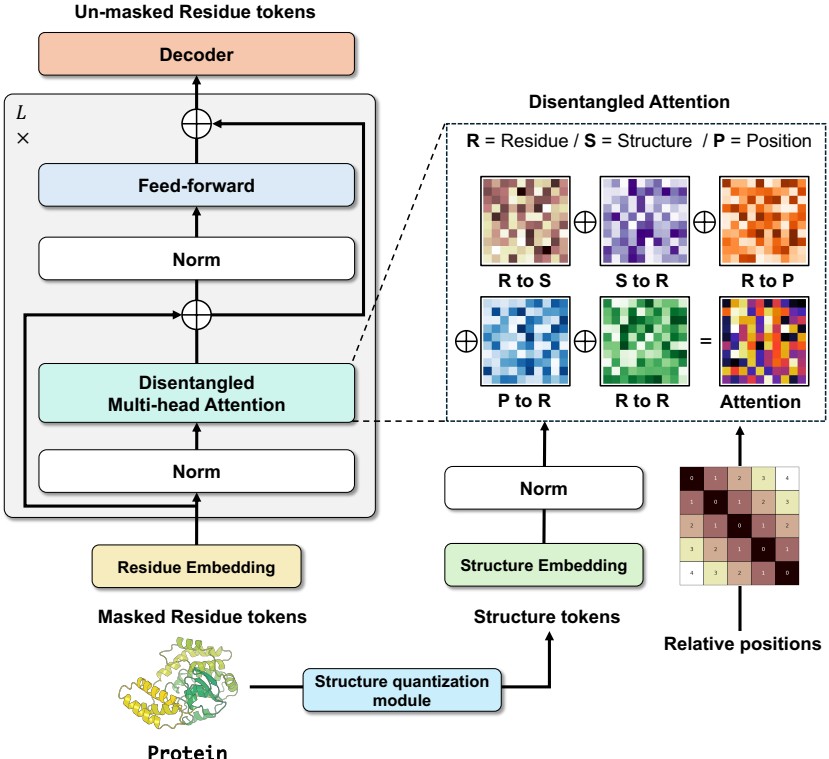

Figure 2: Model architecture of ProSST. ProSST is a Transformer-style model and the difference is that ProSST uses disentangled attention instead of self-attention [19].

Inspired by DeBerta [42], we use an expanded form of disentangled attention to combine the attention of residual sequences and structure sequences as well as relative positions. Specifically, for a residue at position $i$ in a protein sequence, it can be represented by three items: $\boldsymbol{R}_i$ denotes its residue token hidden state, $\boldsymbol{S}_i$ represents the embedding of residue-level local structure, and $\boldsymbol{P}_{i|j}$ is the embedding of relative position with the token at position $j$. The calculation of the cross attention $\boldsymbol{A}_{i,j}$ between residue $i$ and residue $j$ can be decomposed into nine components by:

$$
\begin{aligned}
\boldsymbol{A}_{i,j} &= \{\boldsymbol{R}_i, \boldsymbol{S}_i, \boldsymbol{P}_{i|j}\} \times \{\boldsymbol{R}_j, \boldsymbol{S}_j, \boldsymbol{P}_{j|i}\}^\top \\
&= \boldsymbol{R}_i\boldsymbol{R}_j^\top + \boldsymbol{R}_i\boldsymbol{S}_j^\top + \boldsymbol{R}_i\boldsymbol{P}_{j|i}^\top \\
&\quad + \boldsymbol{S}_i\boldsymbol{R}_j^\top + \boldsymbol{S}_i\boldsymbol{S}_j^\top + \boldsymbol{S}_i\boldsymbol{P}_{j|i}^\top \\
&\quad + \boldsymbol{P}_{i|j}\boldsymbol{R}_j^\top + \boldsymbol{P}_{j|i}\boldsymbol{S}_j^\top + \boldsymbol{P}_{j|i}\boldsymbol{P}_{j|i}^\top.
\end{aligned}
\tag{1}
$$

As formulated in Equation 1, the attention weight of a residue pair can be calculated by separate matrices, including residue tokens, structure tokens, and relative positions. These matrices are utilized for various interactions such as *residue-to-residue, residue-to-structure, residue-to-position, structure-to-residue, structure-to-structure, structure-to-position, position-to-residue, position-to-structure, and position-to-position*. Since our model concentrates on learning contextual embeddings for residues, the terms structure-to-structure ($\boldsymbol{S}_i\boldsymbol{S}_j^\top$), structure-to-position ($\boldsymbol{S}_i\boldsymbol{P}_{j|i}^\top$), position-to-structure ($\boldsymbol{P}_{j|i}\boldsymbol{S}_j^\top$), and position-to-position ($\boldsymbol{P}_{j|i}\boldsymbol{P}_{j|i}^\top$) do not provide relevant information about

residues and thus do not contribute significantly. Consequently, these terms are removed from our implementation of the attention weight calculation. As shown in Figure 2, our sequence-structure disentangled attention mechanism includes 5 types of attention.

In the following part, we use single-head attention as an example to demonstrate the operation of sequence-structure disentangled attention. To begin, we define the relative position of the $i$-th to the $j$-th residue, denoted as $\delta(i, j)$:

$$\delta(i, j) = \begin{cases} 0 & \text{if } i - j \leq -L_{max} \\ 2L_{max} - 1 & \text{if } i - j \geq L_{max} \\ i - j + L_{max} & \text{otherwise} \end{cases}, \tag{2}$$

where, $L_{max}$ represents the maximum relative distance we consider, which is set to $1024$ in the implementation. Similar to standard self-attention operation [19], the computation of query, key for structure, residue and relative position, and the value for residue is as follows:

$$\begin{aligned} \boldsymbol{Q}^r = \boldsymbol{R}\boldsymbol{W}_r^q \quad & \boldsymbol{K}^r = \boldsymbol{R}\boldsymbol{W}_r^k \quad \boldsymbol{V}^r = \boldsymbol{R}\boldsymbol{W}_r^v \\ \boldsymbol{Q}^s = \boldsymbol{S}\boldsymbol{W}_s^q \quad & \boldsymbol{K}^s = \boldsymbol{S}\boldsymbol{W}_s^k \\ \boldsymbol{Q}^p = \boldsymbol{P}\boldsymbol{W}_p^q \quad & \boldsymbol{K}^p = \boldsymbol{S}\boldsymbol{W}_p^k \end{aligned} \tag{3}$$

and the the attention score $\hat{\boldsymbol{A}}_{i,j}$ from residue $i$ to residue $j$ can be calculated as follows:

$$\hat{\boldsymbol{A}}_{i,j} = \underbrace{\boldsymbol{Q}_i^r \boldsymbol{K}_j^{r\top}}_{\text{(a) residue-to-residue}} + \underbrace{\boldsymbol{Q}_i^r \boldsymbol{K}_j^{s\top}}_{\text{(b) residue-to-structure}} + \underbrace{\boldsymbol{Q}_i^r \boldsymbol{K}_{\delta(i,j)}^{p}{}^{\top}}_{\text{(c) residue-to-position}} + \underbrace{\boldsymbol{K}_j^r \boldsymbol{Q}_i^{s\top}}_{\text{(d) structure-to-residue}} + \underbrace{\boldsymbol{K}_j^r \boldsymbol{Q}_{\delta(j,i)}^{p}{}^{\top}}_{\text{(e) position-to-residue}} \tag{4}$$

where $\boldsymbol{Q}_i^r$ represents the $i$-th row of the matrix $\boldsymbol{Q}^r$, and $\boldsymbol{K}_j^r$ denotes the $j$-th row of $\boldsymbol{K}^r$. $\boldsymbol{Q}_i^s$ and $\boldsymbol{K}_j^s$ are the $i$-th and $j$-th rows of $\boldsymbol{Q}^s$ and $\boldsymbol{K}^s$, respectively. The term $\boldsymbol{K}_{\delta(i,j)}^p$ refers to the row in $\boldsymbol{K}^p$ indexed by the relative distance $\delta(i, j)$, and $\boldsymbol{Q}_{\delta(j,i)}^p$ refers to the row in $\boldsymbol{Q}^p$ indexed by the relative distance $\delta(j, i)$. To normalize the attention scores, a scaling factor of $\frac{1}{\sqrt{5d}}$ is applied to $\hat{\boldsymbol{A}}$. This scaling is crucial for ensuring the stability of model training [42], particularly when dealing with large-scale language models. All the $\hat{\boldsymbol{A}}_{ij}$ form the attention matrix, and the final output residue hidden state is $\boldsymbol{R}_o$:

$$\boldsymbol{R}_o = \text{softmax}\left(\frac{\hat{\boldsymbol{A}}}{\sqrt{5d}}\right)\boldsymbol{V}^r, \tag{5}$$

which is used as the input for the hidden state of the next layer.

### 3.3  Pre-Training Objective

ProSST is pre-trained with the structure-conditioned masked language modeling. In this approach, each input sequence $\boldsymbol{x}$ is noised by substituting a fraction of the residues with a special mask token or other residues. The objective of ProSST is to predict the original tokens that have been noise in the input sequence, utilizing both the corrupted sequence and its structure token sequence $\boldsymbol{s}$ as context:

$$\mathcal{L}_{MLM} = \boldsymbol{E}_{\boldsymbol{x}\sim\boldsymbol{X}} \boldsymbol{E}_{\boldsymbol{M}} \sum_{i \in \boldsymbol{M}} -\log p(\boldsymbol{x}_i | \boldsymbol{x}_{/\boldsymbol{M}}, \boldsymbol{s}). \tag{6}$$

We randomly select 15% indices from the set $\boldsymbol{M}$ for nosing and computing loss for back-propagation. At each selected index $i$, there is an 80% chance of substituting the residue with a mask token, a 10% chance of replacing it with a random residue token, and the remaining residues are unchanged. The training objective is to minimize the negative log-likelihood for each noised residue $\boldsymbol{x}_i$, based on the partially noised sequence $\boldsymbol{x}/\boldsymbol{M}$ and the *un-noised* structure tokens, serving as contextual cues. Therefore, to accurately predict the noised tokens, this objective enables the model not only to learn the dependencies between residues but also the relationship between residues and structures. The details of pre-training dataset and hyper-parameter configuration can be found in Appendix A.2.

| Model | Model Type | $\rho_s \uparrow$ | NDCG $\uparrow$ | Top-recall $\uparrow$ |
|---|---|---|---|---|
| EVE [49] | | 0.439 | 0.781 | 0.230 |
| EVmutation [53] | | 0.395 | 0.777 | 0.222 |
| DeepSequence [51] | Evolution-based | 0.407 | 0.774 | 0.225 |
| WaveNet [50] | | 0.373 | 0.761 | 0.203 |
| GEMME [47] | | 0.457 | 0.777 | 0.211 |
| MSA-Transformer [48] | | 0.434 | 0.779 | 0.217 |
| Tranception [21] | | 0.434 | 0.779 | 0.220 |
| RITA [44] | | 0.372 | 0.751 | 0.193 |
| UniRep [45] | | 0.190 | 0.647 | 0.139 |
| ESM-1v [6] | Sequence-based | 0.374 | 0.732 | 0.211 |
| ESM-2 [7] | | 0.414 | 0.747 | 0.217 |
| ProGen2 [22] | | 0.391 | 0.767 | 0.199 |
| VESPA [46] | | 0.394 | 0.759 | 0.201 |
| ESM-IF [37] | Inverse-folding | 0.422 | 0.748 | 0.223 |
| MIF-ST [38] | | 0.401 | 0.765 | 0.226 |
| Trancepiton-EVE [52] | | 0.457 | **0.786** | 0.230 |
| ESM-1v* [6] | Ensemble Models | 0.407 | 0.749 | 0.211 |
| DeepSequence* [51] | | 0.419 | 0.776 | 0.226 |
| SaProt [14] | Sequence-Structure models | 0.457 | 0.768 | 0.233 |
| ProSST | | **0.504** | 0.777 | **0.239** |

Table 1: Comparison of zero-shot mutation prediction performance on ProteinGYM benchmark [43] between ProSST and other models. $\rho_s$ is the Spearman rank correlation.

# 4 Experiments

In this section, we comprehensively evaluate the representation ability of ProSST in several benchmarks, covering zero-shot mutant effective prediction tasks (Section 4.1) and various supervised function prediction tasks (Section 4.2). Additionally, we also provide ablation studies and discussions to further show the effectiveness of the detailed designs in our model (Section 4.3).

## 4.1 Zero-Shot Mutant Effect Prediction

**Datasets**. To evaluate the effectiveness of ProSST in zero-shot mutant effect prediction, we conduct experiments on ProteinGym [43] and utilize AlphaFold2 [11] to generate the structures of wild-type sequences. See Appendix A.2 for the details of the dataset and Appendix A.1 for scoring method.

**Baselines**. We compare ProSST with the current state-of-the-art models, including sequence-based models [6, 7, 21, 44, 45, 22, 46], sequence-structure model [14], inverse folding models [37, 38], evolutionary models [47, 48, 49, 50, 51], and ensemble models [6, 52, 51].

**Results**. Table 1 shows the performance of zero-shot mutant effect prediction on ProteinGYM. Based on the results, we draw several noteworthy conclusions:

- ProSST outperforms all baselines on zero-shot mutant effect predictions of ProteinGYM. We used the non-parametric bootstrap method to calculate the standard error of the difference in Spearman performance between each model and ProSST. The results showed that all standard errors were less than 0.01. This calculation was based on 10,000 bootstrap samples extracted from proteins in the ProteinGym benchmark. Furthermore, ProSST was compared against other models on subsets of ProteinGYM categorized by function, such as stability, activity, binding, and expression. ProSST achieves state-of-the-art (SOTA) performance in the stability, binding, and expression subsets, as detailed in Appendix A.4. Notably, ProSST achieves the best performance in predicting stability, aligning with the previous findings that models incorporating structure information typically perform better in stability predictions [43].

- The degraded version of ProSST (without structure) gets results similar to other sequence-based models. This demonstrates that the performance improvement of our model stems from the efficient modeling of structure information, rather than other factors such as more powerful backbones.

| Model | # Params | DeepLoc Acc% ↑ | Metal Ion Binding Acc% ↑ | Thermostability $\rho_s$ ↑ | GO-MF F1-Max ↑ | GO-BP F1-Max ↑ | GO-CC F1-Max ↑ |
|---|---|---|---|---|---|---|---|
| ESM-2 | 650M | 91.96 | 71.56 | 0.680 | 0.670 | 0.473 | 0.470 |
| ESM-1b | 650M | 92.83 | 73.57 | 0.708 | 0.656 | 0.451 | 0.466 |
| MIF-ST | 643M | 91.76 | 75.08 | 0.694 | 0.633 | 0.375 | 0.322 |
| GearNet | 42M | 89.18 | 71.26 | 0.571 | 0.644 | 0.481 | 0.476 |
| SaProt-35M | 35M | 91.97 | 74.29 | 0.692 | 0.642 | 0.431 | 0.418 |
| SaProt-650M | 650M | 93.55 | 75.75 | 0.724 | **0.682** | 0.486 | 0.479 |
| ESM-GearNet | 690M | 93.55 | 74.11 | 0.651 | 0.676 | **0.516** | **0.507** |
| ProSST | 110M | **94.32**$_{(\pm0.10)}$ | **76.37**$_{(\pm0.02)}$ | **0.726**$_{(\pm0.04)}$ | **0.682**$_{(\pm0.003)}$ | 0.492$_{(\pm0.004)}$ | 0.501$_{(\pm0.002)}$ |

Table 2: Comparison of supervised fine-tuning on downstream tasks. $\rho_s$ denotes the Spearman correlation coefficient.

## 4.2 Supervised Fine-Tuning Tasks

**Downstream tasks.** For supervised learning, we choose four protein downstream tasks, including thermostability prediction, Metal Ion Binding prediction, protein localization prediction (DeepLoc) and GO annotations prediction (three settings including MF, BO, and CC). More details of the tasks, datasets, and metrics can be found in Appendix A.2

**Baselines.** We compared ProSST with other PLMs including ESM-2[7], ESM-1b [5], and the sequence-structure model SaProt [14] (two parameter versions, 35M and 650M), MIF-ST [38], as well as the protein structure representation model GearNet [32] and ESM-GearNet [34].

**Results.** The results of the supervised fine-tuning tasks are shown in Table 4.2, and we can get the following conclusions:

- ProSST gets the best results among all models with 4 firsts in all 6 settings. For the tasks (settings) of DeepLoc, Metal Ion Binding, ProSST largely surpasses other methods, and ESM-GearNet gets comparable (or slightly better) results for thermostability and GO-BP and GO-CC with ProSST, at the price of more than 6× model size.

- The sequence-structure models, ESM-GearNet, SaProt and ProSST, show better results than other counterparts, which suggests the importance of the structure cues in protein modeling. Furthermore, ProSST is more capable of integrating sequence and structure information of proteins than SaProt, which confirms the effectiveness of our designs.

Combined with the results in Section 4.1, ProSST exhibits powerful ability in multiple settings.

## 4.3 Ablation Study

In this section, we provide additional ablation studies and discussions to show the necessity and effectiveness of the detailed designs in ProSST. Specifically, we use zero-shot mutant effect prediction on ProteinGYM, supervised downstream task DeepLoc, and the perplexity in the pre-training validation set to conduct corresponding experiments.

**Ablations on quantized structure.** The ablation results of quantized structure are shown in Table 3 and Figure 3(a), and we can get the following findings:

- We can find, as the increases of $K$ (the size of local structure vocabulary), the performance of ProSST shows an upward trend on all metrics, and most metrics achieve the best results with $K = 2048$. Based on that, we set $K = 2048$ as our default setting.

- As the increase of $K$, the convergence of ProSST improves progressively (Figure 3(a)), which suggests incorporating structure cues can improve the representation capabilities of models.

- Based on the same network architecture, the proposed structure quantization method (with an appropriate hyper-parameter $K$) performs better than Foldseek [16] and DSSP [39], which shows the effectiveness of our design.

| | DeepLoc | ProteinGYM | | | Pretraining |
|---|---|---|---|---|---|
| | Acc% ↑ | $\rho_s$ ↑ | NDCG ↑ | Top-Recall ↑ | Perplexity ↓ |
| ProSST ($K$=4096) | 93.88 (±0.15) | 0.498 | 0.773 | 0.233 | **8.880** |
| ProSST ($K$=2048) | **94.32 (±0.10)** | **0.504** | **0.777** | **0.239** | 9.033 |
| ProSST ($K$=1024) | 93.43 (±0.15) | 0.485 | 0.760 | 0.231 | 9.333 |
| ProSST ($K$=512) | 93.70 (±0.16) | 0.471 | 0.759 | 0.223 | 9.577 |
| ProSST ($K$=128) | 93.14 (±0.04) | 0.469 | 0.753 | 0.228 | 10.021 |
| ProSST ($K$=20) | 93.05 (±0.13) | 0.438 | 0.744 | 0.210 | 10.719 |
| ProSST ($K$=1) | 89.48 (±0.24) | 0.390 | 0.738 | 0.181 | 12.182 |
| ProSST ($K$=0) | 89.77 (±0.26) | 0.392 | 0.741 | 0.184 | 12.190 |
| ProSST (Foldseek) | 93.08 (±0.22) | 0.468 | 0.759 | 0.228 | 10.049 |
| ProSST (DSSP) | 93.16 (±0.16) | 0.439 | 0.760 | 0.204 | 10.009 |

Table 3: Ablation studies on quantized structure. We first show the performance of our models with $K$ centroids of local structures. ProSST ($K$=0) refers to the model without structure token sequence. We also replace the proposed quantization method with existing Foldseek and DSSP, and show the results of these variants.

| | DeepLoc | ProteinGYM | | | Pretraining |
|---|---|---|---|---|---|
| | Acc% ↑ | $\rho_s$ ↑ | NDCG ↑ | Top-Recall ↑ | Perplexity ↓ |
| ProSST | **94.32 (±0.10)** | **0.504** | **0.777** | **0.239** | **9.033** |
| ProSST (- P2R) | 91.31 (±0.14) | 0.478 | 0.778 | 0.227 | 9.173 |
| ProSST (- R2P) | 92.17 (±0.32) | 0.466 | 0.772 | 0.216 | 9.410 |
| ProSST (- R2S) | 90.48 (±0.41) | 0.438 | 0.766 | 0.208 | 12.142 |
| ProSST (- S2R) | 91.27 (±0.20) | 0.475 | 0.779 | 0.226 | 9.355 |
| ProSST (- PE) | 86.05 (±0.65) | 0.095 | 0.634 | 0.126 | 13.885 |
| ProSST (self-attention) | 90.37 (±0.21) | 0.401 | 0.728 | 0.189 | 12.346 |

Table 4: Ablation studies on disentangled attention. The term "-S2R" denotes the removal of structure-to-residue in our attention formulation, similar to other terms, and "- PE" denotes the removal of positional encoding. ProSST (self-attention) refers to the model trained with standard attention (with structure cues).

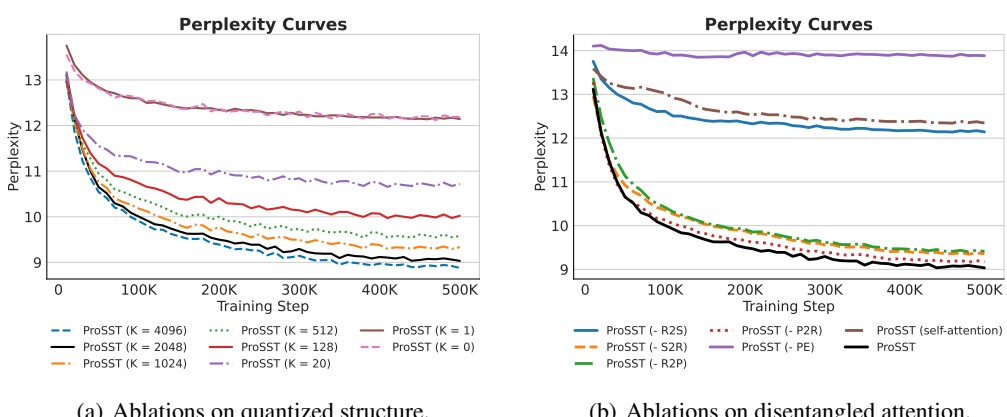

(a) Ablations on quantized structure.    (b) Ablations on disentangled attention.

Figure 3: Perplexity curves of ProSST under different settings. We ablate the components of quantized structure and disentangled attention, and show their perplexity curves on the validation set.

- ProSST (Foldseek), ProSST (DSSP), and all ProSST ($K>0$) models significantly surpass ProSST ($K=0$) in all metrics, which confirms the importance of the structure cues again.

- There is almost no difference in performance between ProSST ($K=1$) and ProSST ($K=0$), indicating that the improvement does not come from the rise in parameters of disentangled attention.

**Ablations on disentangled attention.** Here we show detailed ablations and comparisons of disentangled attention in Table 4 and Figure 3(b), and we can get the following observations:

- All items in Equation 4 are necessary to our attention formulation. Also note that 'P2R' attention has the least impact on model capacity, with the Perplexity slightly increasing from 9.033 to 9.173, suggesting that positional attention to amino acids is relatively less critical than other items. Conversely, removing 'R2S' item results in a significant increase in Perplexity from 9.033 to 12.142, underscoring the important role of structure information in enhancing the model's representation capability.

- Compared with standard self-attention, our attention formulation gets better results for all metrics, indicating that explicitly modeling structure cues is crucial for integrating such information. Besides, positional encoding is also necessary in our design.

As we have mentioned in the Section 2, our disentangled should learn the connections between structure and residue sequence. To valid these, we conduct further experiments to analyze disentangled attention in the Appendix A.5.

## 5   Conclusion and Limitations

This paper introduces ProSST, a protein sequence-structure transformer for PLM. ProSST includes two key techniques, protein structure quantization and sequence-structure disentangled attention. The structure quantization module contains an encoder and a k-means clustering model. The encoder is trained with a denoising objective and is utilized for encoding protein structures. Leveraging this encoder, we embed the local structures of each residue within every protein in the C.A.T.H dataset into a continuous latent space. Then we utilize k-means clustering algorithm to obtain $K$ (default setting is 2048) centroids. These centroids are then utilized to discretize the local structures of residues based on the index of the nearest centroid of its structure embedding vectors. A protein structure can be transformed into a sequence of discrete numbers (or referred to tokens) and each token representing the corresponding local structure of residue. The sequence-structure attention enhances standard self-attention by not only considering self-attention residues but also incorporating attention between residues and structures, and vice versa. This enables the model to learn the relationships between residues and structures, thereby acquiring improved adequate contextual representations of residues. Furthermore, we pre-train ProSST with 18.8 million protein structures using a MLM objective. Experimental results show that ProSST can outperform existing models in ProteinGYM benchmark and other supervised learning tasks. Despite of this, there are some limitations of ProSST. For example, the local structure construction and encoding requires heavy computations. In the future work, we aim to speed up the protein structure quantization process. Another threat is that the structural and sequential data are required for ProSST to derive the final protein representations, since the amount of available structural data is lower than that of sequence data. We provide solutions in the Appendix Section A.6. Additionally, we plan to enhance ProSST by training it with larger structure datasets and expanding its parameter, which may further improve its performance.

## Acknowledgements

This work was supported by the grants from the National Natural Science Foundation of China (Grant Number 12104295), the Computational Biology Key Program of Shanghai Science and Technology Commission (23JS1400600), Shanghai Jiao Tong University Scientific and Technological Innovation Funds (21X010200843), and Science and Technology Innovation Key R&D Program of Chongqing (CSTB2022TIAD-STX0017), the Student Innovation Center at Shanghai Jiao Tong University, and Shanghai Artificial Intelligence Laboratory.

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

# A  Appendix

## A.1  Zero-Shot Scoring

Previous studies have demonstrated that PLMs, when trained on extensive and varied protein sequence databases, are capable of predicting experimental measurements of protein mutants function without further supervision [6, 14]. For those PLMs that are trained with masked language modeling objective, the calculation of mutation scores can be formalized as follows:

$$Score(\boldsymbol{F}) = \sum_{i=1}^{|\boldsymbol{F}|} \log P(\boldsymbol{x}_{p_i} = \boldsymbol{f}_i|\boldsymbol{x}) - \log P(\boldsymbol{x}_{p_i} = \boldsymbol{w}_i|\boldsymbol{x}) \tag{7}$$

Here $\boldsymbol{F}$ is a single or multi-point mutant, and $\boldsymbol{F} = \{(p_i, \boldsymbol{f}_i, \boldsymbol{w}_i)|i = 1, 2, ..., |\boldsymbol{F}|\}$ is a set of triplets, where $p_i \in \mathbb{N}$ represents the mutation position, $\boldsymbol{f}_i$ is the residue after the point mutation, and $\boldsymbol{w}_i$ is the original residue of the point mutation. $\boldsymbol{x}$ is the sequence of residues of the wild type. We slightly modify the formula above to adapt to ProSST, where the structure sequence is an additional condition to score mutants:

$$Score(\boldsymbol{F}) = \sum_{i=1}^{|\boldsymbol{F}|} \log P(\boldsymbol{x}_{p_i} = \boldsymbol{f}_i|\boldsymbol{x}, \boldsymbol{s}) - \log P(\boldsymbol{x}_{p_i} = \boldsymbol{w}_i|\boldsymbol{x}, \boldsymbol{s}) \tag{8}$$

Here, $\boldsymbol{s}$ is the structure token sequence of the wild type.

## A.2  Details of the Datasets and Metrics

**Dataset for pre-training.** The pre-training data is collected from AlphaFoldDB [13], which contains more than 214 million structures predicted by AlphaFold [11]. We downloaded the 90% reduced version, containing 18.8 million structures.[3]. From this collection, we randomly select 100,000 structures for validation (sequences with a similarity of over 30 to the training set will be removed for data deduplication.), enabling us to monitor the perplexity in the training phase. During pre-training, proteins with more than 2048 resiudes (594 samples) are removed for training efficacy.

**Dataset for training structure encoder**. The dataset used for training the structure encoder originates from CATH43-S40 [4]. This dataset is manually annotated and comprises protein crystal structural domains that have been deduplicated for sequence similarity by $40\%$. The original dataset contains 31,885 structures. After removing structural domains missing atoms such as $C\alpha$ and $\mathbf{N}$, the dataset is reduced to 31,270 entries. From this, 200 structures were randomly selected to serve as a validation set. The auto-encoder model was then trained using the configuration that yielded the lowest loss on this validation set.

**Dataset for training structure codebook.** The dataset for training the structure codebook consists of local structures extracted from CATH43-S40. Given a protein structure, slide along the residue sequence to select a segment with a chosen residue as the anchor. Connect up to 40 residues within 10 Å [54] to form a star-shaped graph. As shown in Figure A4, local structures with more than 40 neighbors account for only 0.00052%, indicating that our choice covers most cases. For pairwise amino acid pairs in this graph, if the Euclidean distance is less than 10 Å, a link will be assigned to them. This process yields a number of protein local structures equal to the length of the protein multiplied by the total number of proteins, resulting in 4,735,677 local structures from the protein structures in CATH43-S40. These sub-structures are fed into a structural encoder to obtain embeddings. By setting various quantities for $K$, different structure codebooks are obtained using the k-means clustering algorithm.

**Dataset and metrics for zero-shot mutant effect prediction.**

We utilize the ProteinGYM benchmark [43] to assess the zero-shot mutant effect prediction capabilities of ProSST. ProteinGYM offers comprehensive benchmarks specifically collected for predicting

---

[3]https://cluster.foldseek.com/
[4]http://download.cathdb.info/cath/releases/all-releases/v4_3_0/non-redundant-data-sets/

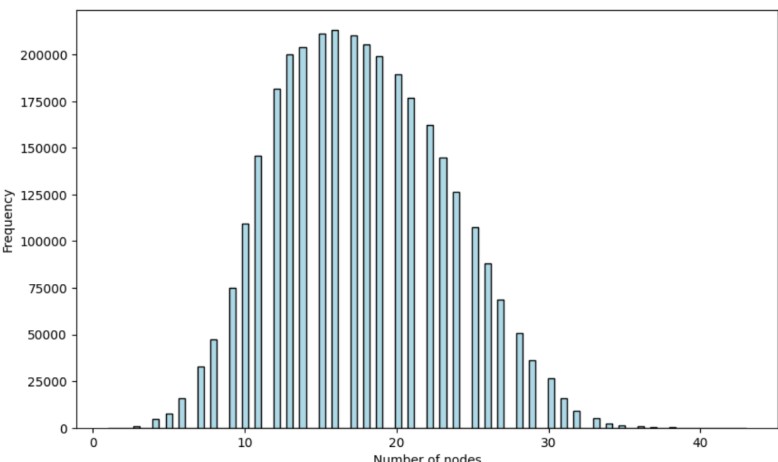

Figure A4: The distribution of the number of residues within 10 Å distance of a local structure.

protein fitness. It contains a wide range of deep mutational scanning assays with millions of mutated sequences. ProSST is evaluated using the most extensively utilized datasets for substitution mutations, which include 217 experimental assays. Each assay incorporates both the sequence and structure of the protein, with a particular emphasis on 66 datasets that focus on thermo-stability. The evaluation metrics employed are the Spearman coefficient, Top-recall, and NDCG, where higher values signify superior model performance. These metrics are computed using scripts [5] provided by ProteinGYM.

**Datasets and metrics for downstream tasks.** The downstream datasets have the same train, valid, and test splits as SaProt's and are downloaded from SaProt. Data statistics are provided in Table A5.

- **Thermostability**. The task is to predict the thermostability values of proteins using the "Human-cell" divisions from the Thermostability task in FLIP [55]. For this regression task, the Spearman correlation coefficient is utilized as the evaluation metric to evaluate the prediction results.

- **DeepLoc** (Protein Sub-cellular Localization). The task is to output a probability distribution across two sub-cellular localization categories for a protein. This is a binary classification task, and we utilize accuracy as the metric to evaluate the predictions. This dataset was introduced by DeepLoc [56] and we use the original data split.

- **Metal Ion Binding**. The task is to predict whether metal ion-binding sites exist within a protein. This is also a binary classification task, and we utilize accuracy as the metric to evaluate the predictions. This dataset was introduced by TAPE [20], and we use the original data split.

- **GO annotations prediction**. This task is to predict Gene Ontology terms to evaluate the model's ability to predict protein functions. This task was introduced by DeepFRI [26], and we use three types of GO labels: Molecular Function (MF), Biological Process (BP), and Cellular Component (CC). This is a multi-label classification task, and we evaluate the model using the Max F1-Score.

### A.3 Details of Implementations

**Structure encoder.** We describe a structure with the graph $G = (V, E)$, adopting the characterizations of $V$ and $E$ as outlined in the GVP framework [18]. The GVP encoder includes a six-layer message-passing graph neural network in which a geometric perceptron replaces the MLP to ensure translational and rotational invariance of the input structure. Our GVP encoder is consistent with the original GVP-GNN [18], except that we removed the residue type information. The GVP encoder is trained from scratch. The dimensions for node and edge representations are set at 256 and 64,

---

[5]https://github.com/OATML-Markslab/ProteinGym/blob/main/scripts/

| Dataset | Training | Valid | Test | Total |
|---|---|---|---|---|
| Termostability | 5,056 | 639 | 1,336 | 7,031 |
| DeepLoc | 5,477 | 1,336 | 1,731 | 8,544 |
| Metal Ion Binding | 5,067 | 662 | 665 | 6,394 |
| Go annotations prediction | 26,224 | 2,904 | 3,350 | 32,478 |

Table A5: Downstream datasets split statistics.

respectively, with the encoder comprising six layers. For optimization, we employ the Adam optimizer in a mini-batch gradient descent approach. To manage computational load, batches are formed by grouping structures of similar sizes, with each batch containing no more than 3000 nodes. The learning rate is set to $1.0 \times 10^{-4}$. The dropout probability is set to $0.01$. And The number of graph layers is set at 6. The training and validation curves of the structure encoder are shown in Figure 5(a).

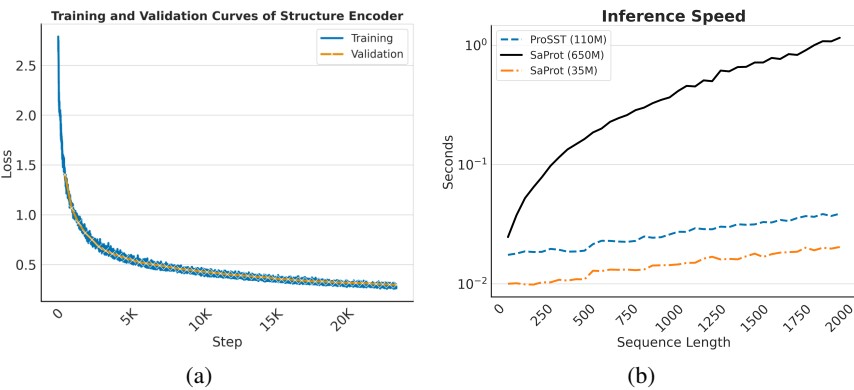

(a)                                         (b)

Figure A5: (a) Training and validation curves of the local-structure auto-encoder. (b) Inference speed of ProSST on different sequence lengths. (batch size = 16).

**Pre-training.** All ProSST models is trained on a DGX-A800 GPU (8×80G) server in BF16 precision for about a month. The model has 12 transformer layers, 12 attention heads, and 768 embedding dims with 3172 feed-forward embedding dimensions with the GELU activation function. We train with 8192 tokens per mini-batch for 500,000 steps. We use AdamW [57] as our optimizer with $\beta_1$ and $\beta_2$ set to 0.9 and 0.999, and a weight decay value of 0.001. We warm up the learning rate from 0 to 0.0002 over the first 2000 steps, then decay it by a cosine schedule to the 0. We use a dropout rate of 0.1 and clip gradients using a clipping value of 1.0. For the tokenization of the protein data, we use the residue-level tokenizer which is adopted in several PLMs [5, 7, 6]. To make the structure sequence the same length as the amino acid sequence, we also added special *[SOS]*, *[EOS]*, and *[PAD]* token for the structure sequences.

**Fine-tuning.** To ensure fair comparisons, we fine-tuned ProSST using a fixed set of hyper-parameters. We use for the Adam optimizer with $\beta_1$ set to 0.9, $\beta_2$ to 0.98, and applied an $L2$ weight decay of 0.001. The batch size was maintained at 64 (If 64 causes the GPU memory to explode, we will reduce the batch size and then use gradient accumulation to achieve the same batch size.) and the learning rate was set at 0.00003, except for Go annotation prediction, where it was adjusted to 0.00001. We fine-tuned all model parameters for 200 epochs, and we choose the best checkpoints based on validation set performance. Following SaProt [14][6], we downloaded all protein structures identified by Uniprot IDs from AFDB [13], and any proteins not found in AFDB were excluded.

**Inference Speed.** We computed the inference speed of ProSST, SaProt (650M) and SaProt (35M) on proteins of different lengths using a batch size of 16 on a server equipped with two Intel 6248R processors and a 3090 GPU and the results are shown in Table 5(b).

---

[6]https://github.com/westlake-repl/SaProt

| Model | Activity | Binding | Expression | Organismal Fitness | Stability |
|---|---|---|---|---|---|
| EVE | 0.464 | 0.386 | 0.408 | 0.447 | 0.491 |
| EVmutation | 0.440 | 0.317 | 0.378 | 0.411 | 0.430 |
| DeepSequence | 0.455 | 0.363 | 0.390 | 0.413 | 0.476 |
| WaveNet | 0.379 | 0.325 | 0.350 | 0.365 | 0.449 |
| GEMME | **0.482** | 0.383 | 0.438 | **0.452** | 0.519 |
| MSA-Transformer | 0.469 | 0.337 | 0.446 | 0.421 | 0.495 |
| Tranception | 0.465 | 0.349 | 0.450 | 0.436 | 0.471 |
| RITA | 0.366 | 0.302 | 0.414 | 0.381 | 0.398 |
| UniRep | 0.182 | 0.202 | 0.216 | 0.141 | 0.210 |
| ESM-1v | 0.396 | 0.268 | 0.405 | 0.362 | 0.437 |
| ESM-2 | 0.425 | 0.337 | 0.415 | 0.369 | 0.523 |
| ProGen2 | 0.402 | 0.302 | 0.418 | 0.387 | 0.445 |
| VESPA | 0.429 | 0.347 | 0.326 | 0.404 | 0.461 |
| ESM-IF | 0.368 | 0.389 | 0.407 | 0.324 | 0.624 |
| MIF-ST | 0.390 | 0.321 | 0.438 | 0.366 | 0.485 |
| Trancepiton-EVE | 0.487 | 0.376 | 0.457 | 0.460 | 0.500 |
| ESM-1v (ensemble) | 0.420 | 0.320 | 0.429 | 0.387 | 0.477 |
| DeepSequence (ensemble) | 0.455 | 0.363 | 0.390 | 0.413 | 0.476 |
| SaProt | 0.458 | 0.378 | 0.488 | 0.367 | 0.592 |
| ProSST | 0.448 | **0.477** | **0.506** | 0.415 | **0.674** |

Table A6: Spearman's rank correlation of baseline models and ProSST on the ProteinGym, separated into five functional categories (Activity, Binding, Organismal Fitness, Stability and Expression).

| Structure token | ProteinGYM ($\rho_s$) | Perplexity |
|---|---|---|
| Original | 0.504 | 9.033 |
| All-zero | 0.112 | 14.524 |
| Random | 0.182 | 14.024 |

Table A7: Performance of ProSST ($K$=2048) using different structure tokens as inputs.

## A.4 Performance of models on the ProteinGYM benchmark separated by functional categories

Table A.4 shows the Spearman's rank correlations on ProteinGYM, categorized by five function types. ProSST achieves state-of-the-art (SOTA) performance in the Stability, Binding, and Expression subsets.

## A.5 Additional experiments on disentangled attention.

We conducted additional experiments to analyze the relationship between disentangled attention.

**Experiment 1.** We replaced all structure tokens in the ProteinGYM and the validation set with zeros or random numbers from a uniform distribution and re-evaluated ProSST. The results are shown in Table A7 The results show that the incorrect structure tokens harmed the performance of ProSST

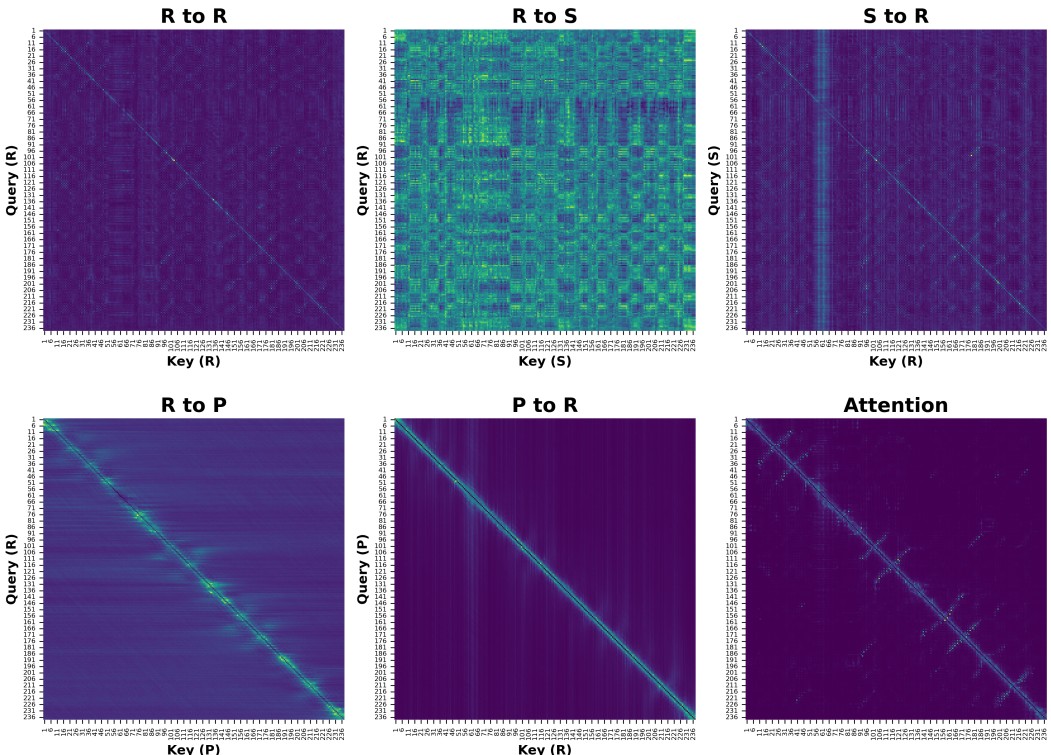

Figure A6: Different types of attentions on Green Fluorescent Protein (GFP). These attentions are the average of each head in the final layer of the Transformer.

| Model | DeepLoc ($Acc\%$) | ProteinGYM ($\rho_s$) | Perplexity |
|---|---|---|---|
| ProSST ($K$=2048) | 94.32 (±0.10) | 0.504 | 9.033 |
| ProSST ($K$=1) | 89.48 (±0.24) | 0.390 | 12.182 |
| ProSST ($K$=0) | 89.77 (±0.26) | 0.392 | 12.190 |

Table A8: Performance comparison of ProSST with special $K$ values.

significantly, suggesting that disentangled attention learned the sequence-structure relationship. Otherwise, the performance would only have a minimal impact.

**Experiment 2.** To verify if disentangled attention can indeed enhance performance, we train ProSST ($K$=1) by substituting the structural tokens with a constant value of 1. This configuration maintains the disentangled attention mechanism even though the structure input is useless. If ProSST ($K$=1) still improves performance, it indicates that the improvement is solely due to the disentangled attention. We evaluate this model on DeepLoc, ProteinGYM and the valid set. The results are shown in Table A8 There is little difference between $K$=1 and $K$=0 since their perplexity curves (refer to Figure 3(a)) nearly overlap. This suggests that disentangled attention alone cannot enhance performance without correct structure tokens.

**Experiment 3.** We visualize the learned different types of attentions on Green Fluorescent Protein (GFP, Unipro ID:P42212 [7]), including 238 residues, in Figure A6. We can see that disentangled attention learns different attention patterns, with notable differences between "R2S" and "S2R".

---

[7]https://www.uniprot.org/uniprotkb/P42212/entry

| Model | Structure Source | ProteinGYM ($\rho_s$) | BLP (Acc%) | Perplexity |
|---|---|---|---|---|
| ProSST ($K$=2048) | AlphaFold2 | 0.504 | 94.32 | 9.033 |
| ProSST ($K$=2048) | ESMFold | 0.471 | 92.73 | 9.144 |
| ProSST (MST) | Missing | 0.438 | 91.84 | 10.325 |
| ProSST (MST) | AlphaFold | 0.456 | 92.31 | 9.447 |
| ProSST ($K$=0) | Missing | 0.392 | 89.65 | 12.190 |

Table A9: Performance comparison of ProSST with different special $K$.

We can conclude from Experiment 1 and Experiment 2 that the disentangled attention and structure tokens sequence are inseparable. The disentangled attention mechanism cannot function without correct structure tokens. This also indicates that the performance improvement of our model stems from the design of the model rather than increasing its parameters of the attention layer. Furthermore, the Experiment 3 shows that our disentangled attention actually learned different patterns of attentions.

### A.6 Solutions to Sequence-only Datasets

In conclusion, we offer two solutions for obtain representations of the sequence-only protein datasets:

- Utilize AlphaFold 2 [11] or ESMFold [7][8] for structure prediction as they are highly reliable methods.
- Use ProSST (MST), which is trained with structure masking and supporting sequence-only inputs. The MST denotes "Masked Structure Training (MST)", which means that during pre-training, each sample's structure sequence has a 50% probability of being replaced by a fully masked sequence [1,1,1,1,1,...,1], simulating missing protein structure. Therefore, when applying ProSST to sequence-only datasets, we need to use the masked sequence [1,1,1,1,1,...,1] as a substitute for the structure token sequence.

We have evaluated the two approaches on the ProteinGym benchmark, binary localization prediction (BLP) from a sequence-only benchmark, PEER [58], and perplexity on the validation set. The results are show in Table A9: In the Table A9, the first two rows show the performance differences between AlphaFold and ESMFold. Rows 3-4 show the performance of the new model ProSST(MST). And row 5 shows the performance of the sequence-only model.

### A.7 AlphaFold pLDDT versus Zero-shot mutant effect performance

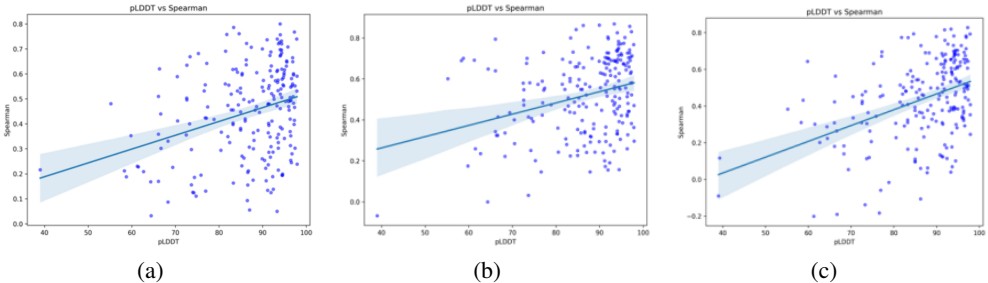

|  (a)  |  (b)  |  (c)  |

Figure A7: (a) pLDDT vs. Spearman of ProSST on ProteinGYM. (b) pLDDT vs. Spearman of SaProt on ProteinGYM. (c) pLDDT vs. Spearman of ESM-IF1 on ProteinGYM.

Protein structures containing disorder region may not be accurately predicted by AlphaFold 2, potentially leading to reduced performance of structure-aware models. Here, we test the relationship

---

[8]https://esmatlas.com/resources?action=fold

between AlphaFold pLDDT scores and the performance of structure-aware models including ProSST, SaProt, and ESM-IF on ProteinGYM, as illustrated in Figures A7. Our findings reveal a positive correlation between pLDDT values and model performance: a correlation coefficient of 0.30 for ProSST, 0.31 for SaProt, and 0.42 for ESM-IF1, where the correlation coefficient ($\rho_p$) represents the strength of the relationship. These results suggest that structure-aware models may exhibit limitations in accurately predicting the structures of disordered proteins.

