# OpenReview forum: "ProSST: Protein Language Modeling with Quantized Structure and Disentangled Attention"
_NeurIPS.cc/2024/Conference — NeurIPS 2024 poster_

### Official Review · Reviewer_jE9F · 2024-07-12

**Soundness:** 3
**Presentation:** 3
**Contribution:** 3
**Rating:** 7
**Confidence:** 4

**Summary:**

In this paper the authors introduce ProSST, a method for training protein language models on both sequence and structure. They use a geometric vector perceptron trained with a structural denoising objective to obtain a structure encoder, and then perform k-means clustering on the embeddings of local residue neighborhoods in CATH to obtain a codebook of structural tokens. A masked language model is trained with disentangled attention, conditioned on the structural token inputs and the corrupted sequence.

**Strengths:**

The approach follows a clearly promising and useful trend of incorporating explicit structural information into protein language models. ProSST improves on prior work by expanding the codebook size of structural tokenizers and showing the effects of disentangled attention for mutation effect prediction and improved pre-training perplexity. The authors ablate their design decisions to give evidence that these two changes are indeed improvements on traditional pLM design. The results on ProteinGym seem to significantly improve on pLMs without structural information, and even show noticeable improvements over SaProt.

**Weaknesses:**

It is not clear why the structural quantization is a two-step process, involving training a separate structure encoder with a denoising objective, and then clustering to form the local codebook. An advantage of SaProt is the simplicity of the codebook and almost negligible cost of structural tokenization. How does this compare to the pre-processing needed for ProSST?

**Questions:**

Can the authors give some indication about the increased time and computational cost of tokenizing structures using their approach?

Why is the two-step structural quantization approach used, rather than a typical VQ-VAE as used in FoldSeek? Are there advantages to training stability, simplicity, or inference time tokenization?

Have the authors investigated using smaller neighborhood sizes than 40 nearest neighbors? Is this a route to reducing the computational burden of structural quantization?

**Limitations:**

Yes

---

> ### Author Rebuttal · Authors · 2024-08-07
>
> Thank you very much for recognizing the novelty and contribution of our work. Your insightful comments helped us enrich the analysis a lot. With the response below, we hope to address your concerns properly.
>
> **Weakness:**
>
> **W1.** In fact, we do not need to train the structure encoder and the clustering model for every protein. Our structure quantization serves as a data preprocessing step. The codebook is only pre-trained on a pre-training structure encoder dataset (Appendix A.2, Dataset for training structure codebook). SaProt uses FoldSeek [1] as its structure codebook, which also needs pre-training. Since training a codebook is FoldSeek's task, it isn't explicitly detailed in SaProt's article. The primary difference between our structure quantization module and FoldSeek is that our local structure considers up to the nearest 40 residues within a 10 Å distance, whereas FoldSeek only incorporates structure information between preceding and succeeding residues.
>
> **Q1.**  Tokenizing a protein structure containing $L$ residues involves three steps:
>
> **Step 1.** For  $i = 1, 2, 3, \ldots, L$, generate a local structure for residue $r_i$, converting it into a graph $G_i$. (Left side of Figure 2c)
>
> **Step 2.** For  $i = 1, 2, 3, \ldots, L$, encode each graph $G_i$ into a continuous vector $e_i$  using  the trained structure encoder.
>
> **Step 3.** For  $i = 1, 2, 3, \ldots, L$, utilize the trained clustering codebook to convert $e_i$ to a token $s_i$.
>
> We also evaluated the time required to convert a protein into structure tokens for five proteins of different lengths using a server equipped with an RTX 1080 GPU and a Intel Xeon E5-2690 CPU. Step 2 was executed on the GPU, while steps 1 and 3 were executed on the CPU. The results are as follows.
>
> | Protein Name (Uniprot_ID) | Length (Local structures) | Step 1 | Step 2 + Step 3 |
> | --- | --- | --- | --- |
> | CCDB_ECOLI_Adkar_2012 | 101 | 0.29s | 4.43s |
> | ESTA_BACSU_Nutschel_2020 | 212 | 0.67s | 4.27s |
> | PTEN_HUMAN_Matreyek_2021 | 403 | 1.06s | 4.45s |
> | ENV_HV1B9_DuenasDecamp_2016 | 853 | 3.24s | 5.63s |
>
> We believe the speed is acceptable. Nevertheless, one of our important future tasks is to use parallelization techniques to optimize the speed of subgraph segmentation (Step 1) and local structure encoding (Step 2). We are working on this development and will release it in the camera-ready version.
>
> **Q2.** Both our structure quantization module and FoldSeek involve two steps: training a structure encoder and a clustering model. We chose the denoising autoencoder for its focus on representation learning and simplicity, which better suits our needs. While VAE and VQ-VAE are generative models, we do not require a structure generation model. The training curve of our structure encoder is shown in **Figure R9**. The change of training loss and validation loss is stable.
>
> **Q3.** Smaller neighborhood sizes may ignore some important neighbors. Although using neighborhood sizes is a route to accelerate average pooling, it may mislead some neighboring nodes. We selected the nearest **40** residues within a distance of **10 Å**, according to [4]. As shown in Figure R1, local structures with more than **40** neighbors account for only 0.00052% (5.2e-6). Thus, 40 can cover most cases.
>
> References
>
> [1] Clustering predicted structures at the scale of the known protein universe.
>
> [2] Convolutions are competitive with transformers for protein sequence pretraining.
>
> [3] https://www.uniprot.org/uniprotkb/statistics
>
> [4] Discovery of Novel Gain-of-Function Mutations Guided by Structure-Based Deep Learning.

---

> > ### Comment · Reviewer_jE9F · 2024-08-12
> >
> > I thank the authors for their response and will maintain my score.

---

> > > ### Author Response · Authors · 2024-08-13
> > >
> > > We deeply appreciate the dedication you have shown in evaluating our submission and offering your comments. Your insightful recommendations have played a crucial role in improving the clarity and rigor of our work. Regardless of the ultimate outcome, your input has enhanced the clarity and quality of our work. We extend our heartfelt thanks to you!

---

### Official Review · Reviewer_1pAx · 2024-07-13

**Soundness:** 3
**Presentation:** 2
**Contribution:** 2
**Rating:** 5
**Confidence:** 4

**Summary:**

This paper focuses on the protein representation task. ProSST introduces a quantized method to combine the information from protein structure. Additionally, the authors also propose a disentangled attention mechanism on top of the quantized structures to learn the relationship between residue and structure token sequences. They claim that the proposed method outperforms state-of-the-art methods in several tasks under both zero-shot and supervised learning settings.

**Strengths:**

1. The paper is easy to follow and the presentation is clear.
2. The authors proposed an interesting approach to merge representations of two protein language models based on sequences and structures. The authors provide extensive experiments to confirm the effectiveness of the proposed methods.
3. While cross-attention between sequence and structure has been widely explored before, the proposed disentangled attention seems novel. The introduction of the relative position encoding also seems beneficial according to the ablation.

**Weaknesses:**

1. The novelty is somewhat limited. I do believe such work is meaningful for computational biology, but this is not the first hybrid approach to protein language models (such as ESM-GearNet [1]).
2. Regarding the experiment, I strongly suggest comparing ProSST to ESM-GearNet, which I think is a fairer comparison considering the utilization of both sequence and structure.
3. Besides comparing the number of parameters, I suggest the authors provide experiments on inference speed. In my experience, clustering is often time-consuming, especially for long sequences and multiple cluster centers (K > 2000 in this paper).

[1] A Systematic Study of Joint Representation Learning on Protein Sequences and Structures

**Questions:**

1. How does ProSST deal with extremely long protein sequences?
2. How to deploy this method to sequence-only datasets like the PEER benchmark [2]? It seems ProSST requires both sequence and structure as inputs.

[2] Peer: a comprehensive and multi-task benchmark for protein sequence understanding

**Limitations:**

My main concern is the experiment setting mentioned in the Weaknesses part. Authors are welcome to answer my questions and I am leaning to raise my rate.

---

> ### Author Rebuttal · Authors · 2024-08-07
>
> We sincerely appreciate your thoughtful feedback on our work and provide a thorough response addressing your main concerns.
>
> **Weakness**
>
> **W1.**  We agree that our work is not the first hybrid approach to protein language models, and we have cited works such as ESM-GearNet and LM-GVP in Section 2.1. However, what sets our work apart is our disentangled attention mechanisms and structure quantization module. We redesigned the self-attention mechanism in the Transformer to accommodate multiple sequence inputs, including protein sequence, structure sequence, and relative position matrix sequence. Meanwhile, the structure quantization module can encode local structure of residues with more comprehensive information. Evaluation results demonstrate the efficacy of our model across various protein understanding tasks, especially in zero-shot mutant fitness prediction. We contend that the overall model, local structure encoding and quantization, the disentangled attention mechanism, and the evaluation results jointly represent a valuable contribution to the field of computational biology.
>
> **W2.** We evaluated ESM-GearNet on the fine-tuning task and compared it to ProSST.  The results are as follows:
>
> |  | DeepLoc | Metal Ion Binding | Thermostability | GO-MF | GO-BP | GO-CC |
> | --- | --- | --- | --- | --- | --- | --- |
> | ProSST | 94.32(±0.10) | 76.37(±0.02) | 0.726(±0.04) | 0.682 (±0.003) | 0.492 (±0.004) | 0.504 (±0.002) |
> | ESM-GearNet | 93.55 | 74.11 | 0.651 | 0.676 | 0.516 | 0.507 |
>
>
> The scores of ESM-GearNet are derived from SaProt. Note that the slightly different scores compared to the manuscript are due to the updated random seed and dataset. We have re-evaluated ProSST with different seeds in order to compute the average performance. Additionally, the evaluation was conducted on the updated GO dataset by SaProt.
>
> **W3.** The clustering model is trained only once. The training and inference speeds of the clustering model are as follows:
>
> | K | Clustering Model Training Time | Clustering Model Inference Time |
> | --- | --- | --- |
> | 20 | 5m 22s | 5 ms |
> | 128 | 9m 39s | 5 ms |
> | 512 | 26m 41s | 8 ms |
> | 1024 | 51m 16s | 15 ms |
> | 2048 | 98m 4s | 26 ms |
> | 4096 | 185m 50s | 250 ms |
>
> We also show the inference time of ProSST(110M), SaProt(35M), SaProt(650M) in **Figure R8**. ProSST(110M) is faster than SaProt (650M) and slower than SaProt (35M). The experiments are conducted on a server with one RTX 3090 GPU and two Intel 8468 CPUs. We will discuss inference speed in revised paper.
>
> **Questions**
>
> **Q1.** ProSST utilizes relative positional encoding, which supports inference without constraints on sequence length. For pre-training, we removed proteins longer than 2048 residues for training efficiency (extremely long sequences may cause an overflow of CUDA memory). However, we do not do any truncation during fine-tuning and zero-shot mutant effect prediction. Furthermore, we have evaluated the perplexity of ProSST on a long protein dataset containing 594 proteins longer than 2048 residues. The statistics of the sequence lengths and evaluation results are as follows:
>
> | Count | Length Mean. | Length Min. | Length Max. | Length Std. | Perplexity |
> | --- | --- | --- | --- | --- | --- |
> | 594 | 2313 | 2049 | 2699 | 180 | 9.013 |
>
> The perplexity is **9.013**, which is similar to the validation set perplexity of **9.033**, suggesting that ProSST can effectively understand long protein sequences. Another notable point is that extremely long protein sequences are very scarce in the nature, as almost **99.7%** of proteins are shorter than 2048 [1]. We will add the discussion about long sequences to the manuscript.
>
> **Q2.:** The current implementation requires to input both sequence and structure. Extending the framework to accept sequence-only datasets is possible. Here, we provide three approaches:
>
> **Approach 1:** Use AlphaFold or ESMFold to predict structures.
>
> **Approach 2:** Use ProSST(MST), trained with structure masking and supporting sequence-only inputs. We will release a model ProSST(MST), an extension of the ProSST model that incorporates Masked Structure Training (MST). The MST means that during pre-training, each sample's structure sequence has a 50% probability of being replaced by a fully masked sequence [1,1,1,1,1,…,1], simulating missing protein structure. Therefore, when applying ProSST to sequence-only datasets, we need to use the masked sequence [1,1,1,1,1,…,1] as a substitute for the structure token sequence.
>
> We have evaluated these methods on the ProteinGym benchmark, binary localization prediction (BLP) from the PEER benchmark, and perplexity on the validation set. The results are as follows:
>
> | Model | Structure Source | ProteinGym | BLP(Peer) | Perplexity |
> | --- | --- | --- | --- | --- |
> | ProSST (K=2048) | AlphaFold2 | 0.504 | 94.32 | 9.033 |
> | ProSST (K=2048) | ESMFold | 0.471 | 92.73 | 9.144 |
> | ProSST (MST) | Missing | 0.438 | 91.84 | 10.325 |
> | ProSST (MST) | AlphaFold | 0.456 | 92.31 | 9.447 |
> | ProSST (K=0) | Missing | 0.392 | 89.65 | 12.190 |
>
> Rows 1-2 show the performance differences between AlphaFold and ESMFold. Rows 3-4 show the performance of the new model ProSST(MST). Row 5 shows the performance of the sequence-only model. A discussion on how to deploy ProSST on sequence-only datasets will be added to the revised version.
>
> Reference
>
> [1] https://www.uniprot.org/uniprotkb/statistics

---

> > ### Comment · Reviewer_1pAx · 2024-08-11
> >
> > Thank you for the responses, I believe my concerns have mostly been addressed, although I still think there exists many similarities with previous work. I do think that this is now above the acceptance threshold and have changed my score.

---

> > > ### Author Response · Authors · 2024-08-13
> > >
> > > We thank you for taking the time and effort to review our research and engage with our responses. Your comments have helped us refine our paper. Regardless of the final outcome, your feedback has helped us better define and present our research objectives.

---

### Official Review · Reviewer_cgxa · 2024-07-14

**Soundness:** 3
**Presentation:** 3
**Contribution:** 2
**Rating:** 7
**Confidence:** 5

**Summary:**

The paper proposes ProSST (Protein Structure-Sequence Transformer), a novel PLM incorporating sequence and structure information. ProSST can be split into two parts: a modified version of the Transformer architecture and a quantization module. The quantization module consists of a structure encoder using a Geometric Vector Perceptron (GVP). The authors train the GVP on a large dataset of protein structures (18.8 million) in which they encode residues by contextualizing local structures with surrounding ones in a manner more robust than current methods. They then discretize local residues into tokens using a pre-trained k-means clustering model. ProSST adds relative positional information to the structure before feeding it into the Transformer architecture. Instead of standard self-attention, the authors propose using disentangled attention, a method that allows the model to process contextual information from sequence and structure tokens. They test their model on various downstream protein function tasks, for which they achieve state-of-the-art performance across almost all metrics.
The main contributions of the paper are as follows:
● Introduced a novel method for protein structure quantization by incorporating contextual structural information into a PLM using a structure autoencoder
● Proposed a nonstandard attention calculation that allows for sequence and structure information to be incorporated into protein function prediction

**Strengths:**

- The paper clearly defines their contributions at the beginning of the paper and justifies most of the design choices through a proper literature review.
- The paper presents current gaps in research and proposes relevant methods to bridge this gap.
- The methods section provides sufficient design details, allowing for the model architecture to be reproduced.
- Similarly, the formulation for both disentangled attention and the training objective are well defined.
- The tasks outlined in the experiments section range in variety, covering many different facets of protein function prediction.
- Following this, the ablation study provided by the authors is robust in its analysis of architecture choices, justifying empirically the choices that are made.
- Overall, some of the claims made in the introduction are justified by the experimental results.

**Weaknesses:**

- Despite having clearly formulated reasons for choosing their quantization methods, the authors need a similar justification for disentangled attention. There is little discussion on this design choice until after the related works section. Similarly, there is little experimental analysis of disentangled attention outside the ablation study. This is surprising as the departure from standard self-attention allows for a nearly four percent increase in performance, which is just as impactful as their quantization module. The paper talks about learning the connections between structural tokens and sequence tokens as the main contribution of ProSST but never demonstrates the connection.

- The paper conveys the generalization power of their model by testing it on multiple downstream protein function prediction tasks. ProSST outperforms nearly every other model on these benchmarks, yet some experimental setups raise concerns. Most glaringly, the paper provides little information on their testing regimen across all training tasks. In the experiments that do outline testing, the authors describe extremely small validation sets. For example, in the training phase, the authors train ProSST on 18.8 million samples, with 100,000 being used for validation (~ 99:1 split). Similarly, the split for training the structure codebook is 31,070 train points with only 200 points for validation (~ 99:1 split). The paper needs to provide more information on the testing regimen for the fine-tuning tasks. Along with omitting these training details, the authors seemingly make many mistakes in reporting their results. First and foremost, the paper provides no error or significance analysis despite including empirical evaluations of ProSST (which is required in the NeurIPS paper checklist).

- In Table 3 and Table 4 on page 8, the paper evaluates the effectiveness of ProSST, marking the best results in bold. When comparing the best results from both tables, it is clear that the ablation study was done with K=2048 being their clustering hyperparameter due to the matching accuracies (although this is not reported explicitly). However, almost all of the other metrics outside of accuracy for this experiment do not match, which is either a copying mistake by the authors or is indicative of a different experiment. If the latter explanation is true, then the variance in the results indicates that the paper should have been more thorough in its analysis of the model’s performance across initializations and data splits. Along with the extremely small validation set sizes and little testing information, this makes me skeptical of the empirical results section. In addition, in Table 1, the authors incorrectly bold their model’s performance on the NDCG metric when it is clear that five other models outperform theirs on this specific metric.

Another area for improvement of the paper is its analysis of the structure quantization module. The model has a four percent increase in performance when jumping from K=0 to K=20 and has diminishing returns from there onward. At K=0, there are no structure tokens passed to the model, meaning that disentangled attention is unable to work effectively in this experimental setting. This becomes an issue when you compare the results to the ablation study on disentangled attention; it is difficult to compare the effects of the quantization module to disentangled attention. Because of this, their results might suggest that the increase in performance is due to disentangled attention, not necessarily their quantization module.

The conclusion section is also a weakness of thenot give much insight into the methods proposed and instead reviews the methods section. It lacks a thorough ana paper. It does lysis of the paper's results.

**Questions:**

Is there a reason why several of the baselines (e.g., EVmutation, DeepSequence, WaveNet, UniRep, RITA, ProGen2, and VESPA) and datasets (DMS data other than thermostability) from ProteinGym are not included in this study?

**Limitations:**

- Contribution is limited given that now ESM3 and many other PLMs consider structure in the language model
- There has been no discussion on several protein functions that do not benefit from rigid 3D structure, as predicted by AF2/3, e.g., disordered protein functions with floppy regions: how would ProSST handle those cases?

Conclusion:

This paper provides interesting methods for improving PLM performance by intelligently incorporating structural information into protein function prediction problems. The two novelties of this paper come from their quantization method and disentangled attention application. Despite providing compelling methods and promising results, there are many details that are glossed over, which over the course of the paper accrue many concerns from the reader on the validity of the results. My suggestion to the authors is to provide more insight into their choice of disentangled attention (related works section) and analysis of the results (conclusion). Similarly, the experiment section and supplemental materials need to be updated with enough information for the reader to faithfully reproduce the results according to best practices. In general, there is enough concern from the experimental results that warrant justification and response from the authors.

---

> ### Author Rebuttal · Authors · 2024-08-07
>
> Thank you for your meticulous feedback. Below are our responses.
>
> **Weakness:**
>
> **W1.**  We conducted additional experiments to analyze disentangled attention.
>
> **Experiment 1**: We replaced all structure tokens in the test set with zeros or random numbers from a uniform distribution and re-evaluated ProSST. The results are:
>
> |Structure|ProteinGym|Perplexity|
> |---|---|---|
> | Original|0.504|9.033|
> | All-zero|0.112|14.524|
> | Random|0.182|14.024|
>
> Incorrect structure tokens decreased performance, suggesting that **disentangled attention learned the sequence-structure relationship.** Otherwise, performance would have been less affected.
>
> **Experiment 2**: We train ProSST (K=1), where the structure tokens are replaced with a constant value of 1. This setup helps preserve the disentangled attention mechanism. If ProSST (K=1) still improves performance, it indicates that the improvement is solely due to the disentangled attention.
>
> | |DeepLoc|ProteinGym|Perplexity|
> |---|---|---|---|
> |ProSST(K=2048)|94.32(±0.10)|0.504|9.033|
> |ProSST(K=1)|89.48 (±0.24)|0.390|12.182|
> |ProSST(K=0)|89.77 (±0.26)|0.392|12.190|
>
> There is little difference between K=1 and K=0, as their perplexity curves (see **Figure R3**) almost overlap. This indicates that **disentangled attention cannot improve performance without correct structure tokens**.
>
> **Experiment 3:**  Visualizing the learned attentions. We show different types of attentions on Green Fluorescent Protein (GFP), including 238 residues, in **Figure R4**. We can see that **disentangled attention learns different attention patterns**, with notable differences between R2S and S2R.
>
> **W2.**
>
> (1) **Dataset Split**
>
> (1.1) There is no test set for Pre-training for the structure codebook and Transformer. We use only a small validation set (without a test set) for pre-training to maximize data usage, similar to pre-trained models like ESM-2, which use 0.5% of data for validation [1].
>
> (1.2) The fine-tuning datasets have train/valid/test splits like SaProt's and are downloaded from SaProt. Data statistics will be provided in the revised paper.
>
> (1.3) In this zero-shot mutant effect prediction, all data are in the test set.
>
> **(2) Error and Significance Analysis**
>
> (2.1) We will add error analysis to fine-tuning datasets. We use the same hyperparameters and repeated experiments five times with different seeds. The average performance served as the metric, with the standard deviation as the error. The results are:
>
> | |DeepLoc|Metal Ion Binding|Thermostability|GO-MF|GO-BP| GO-CC |
> |---|---|---|---|---|---|---|
> |ProSST|94.32(±0.10)|76.37(±0.02)|0.726(±0.04)|0.682(±0.003)|0.492(±0.004)|0.504(±0.002)|
>
> Note that the slightly different scores compared to the manuscript are due to the updated random seed. Additionally, the evaluation was conducted on the updated GO dataset by SaProt.
>
> (2.2) The error in the ablation study tables will be updated in the revised version, which would not change the performance ranking.
>
> (2.3) We used the non-parametric bootstrap method for zero-shot mutant fitness prediction to test differences between the baselines and ProSST. In all tests, the p-values were less than 0.01.
>
> **W3.** The inconsistency between Table 3 and Table 4 was due to a copying error. The data for ProSST (K=2048) in Table 3 is correct, but an error occurred when copying to Table 4. The performance of ProSST in Table 4 on ProteinGym should be 0.504 for Spearman, 0.777 for NDCG, and 0.239 for Top-recall. We will correct these and fix the black marking issues in all tables.
>
> **W4.** Disentangled attention cannot improve performance without correct structure tokens. We train ProSST (K=1), where the structure tokens are replaced with a constant value of 1. This setup helps preserve the disentangled attention mechanism. If ProSST (K=1) still improves performance, it indicates that the improvement is solely due to the disentangled attention. However, ProSST(K=1) is not better than ProSST(K=0). The results can be referred to response to **W1**.
>
> **W5.** We will emphasize the reasons for selecting discrete structure tokens and using disentangled attention, and we will highlight the analysis of the relationship between them, as discussed in W1.
>
> **Questions:**
>
> **Q1.** We select baselines with high Spearman correlations from the ProteinGym webpage [2]. We also compared the mentioned baselines to ProSST and will update it in our paper. The results are as follows:
>
> |Model|ρs|NDCG|Top-recall|
> |---|---|---|---|
> |EVmutation|0.395|0.777|0.222|
> |DeepSequence|0.407|0.774|0.225|
> |WaveNet|0.373|0.761|0.203|
> |RITA|0.372|0.751|0.193|
> |UniRep|0.19|0.647|0.139|
> |ProGen2|0.391|0.767|0.199|
> |VESPA|0.394|0.759|0.201|
> |ProSST|0.504|0.777|0.239|
>
> Since structure models often excel in the Stability subset, we additionally compared ProSST with other models in this subset. We will include other ProteinGYM subsets (Stability, Activity, Binding, Expression) in the Appendix. ProSST achieves state-of-the-art (SOTA) performance in the Stability, Binding, and Expression subsets.
>
> **Limitations:**
>
> **L1.** We believe our model is a valuable addition to structure-aware protein language models. Note that ESM-3 was introduced in June 2024, which was after our submission to NeurIPS in May 2024.
>
> **L2.** We utilize AlphaFold to predict structures of disordered proteins. We provide the relationship between AlphaFold pLDDT and the performance of structure models like ProSST, SaProt, and ESM-IF on ProteinGYM, as shown in **Figure R5-R7**. There is a positive correlation between pLDDT and model performance: for ProSST, $\rho=0.30$; for SaProt, $\rho=0.31$; and for ESM-IF1 $\rho=0.42$, where $\rho$ is the Pearson correlation coefficient. This may indicate that structure models may not perform well on disordered proteins.
>
> References
>
> [1] Evolutionary-scale prediction of atomic-level protein structure with a language model.
>
> [2] https://proteingym.org/benchmarks

---

> > ### Comment · Reviewer_cgxa · 2024-08-11
> > **Thanks for the response**
> >
> > My concerns are mostly addressed. I increase my score.

---

> > > ### Author Response · Authors · 2024-08-13
> > >
> > > We are truly grateful for the time and effort you dedicated to reviewing our work and considering our responses. Your constructive feedback and expert advice have significantly enhanced the quality of our research. Regardless of the final decision, your thoughtful engagement has greatly clarified the objectives and structure of our paper. Thank you for your invaluable contribution!

---

### Official Review · Reviewer_t3Vw · 2024-07-19

**Soundness:** 3
**Presentation:** 4
**Contribution:** 3
**Rating:** 7
**Confidence:** 4

**Summary:**

This paper presents ProSST, a new language model for protein data that captures both structural and sequential modalities of proteins. The protein's structure is tokenized using a graph-based auto-encoder architecture, where each residue's local structure is embedded into a vector in a high-dimensional embedding space and then classified using a $k$-means clustering algorithm within that embedding space. The two sequence-based and structure-based data streams are then passed to multiple disentangled attention layers, allowing the model to better capture the connections between structure and sequence data. Numerical results showcase the outperformance of ProSST over several baselines across a range of downstream tasks.

**Strengths:**

- Fusing protein structure and sequence data is an important and timely research topic, and the proposed architecture provides a novel way of combining these two modalities to derive more expressive protein representations.

- The numerical results and ablation studies are quite comprehensive and show considerable gains over state-of-the-art benchmarks.

- The paper is very well written and easy to follow for the most part.

**Weaknesses:**

The main weakness of this work, in my opinion, is that both structural and sequential data are required for the model to derive the final protein representations, since the amount of available structural data is lower than that of sequence data. I wonder if there is any way in which the model can operate on sequence input alone, with the structure knowledge embedded into the model parameters. Especially, how does the model work on protein sequences for which no structural data is available? Is there a "sequence-only" mode that the model can revert to (e.g., by feeding a "default/noise" structure input alongside the actual sequence)? I see the ProSST (-structure) model in Table 1, but that seems to be a model that was only trained on sequence data, so the training pipeline and its parameterization are completely different than the complete ProSST framework.

**Questions:**

- What is the reasoning behind choosing **40** residues as the neighborhood of each node in Section 3.1?

- Is $L$ on line 128 the same as $l$ on line 137?

- Could you please elaborate on how the GVP encoder is parameterized?

- Have you studied what would happen if, instead of discrete structure tokens on the bottom-right of Figure 1, the centroid embedding of each cluster is used as the input structural vector? Also, what if the continuous local structure embeddings are directly used and the clustering is removed altogether?

- Could you explain the average pooling that happens on line 146? In particular, what does $l$ precisely represent here? Is it the 40-node neighborhood of each residue's local graph?

- What is the difference between the disentangled attention in Eq. (1) and a regular attention mechanism, where the tokenized representations of each residue are simply concatenated together?

- What is $k$ in Eq. (2)? Is it the same as the number of clusters in $k$-means clustering of structural embeddings?

- Why is the structure $s$ in Eq. (6) left unmasked? Shouldn't you also mask $s$ when you make the residue tokens in the sequence data? On a related note, shouldn't the structure of the variant sequence be changed in the first term of Eq. (8) when calculating the scores?

- It would be helpful if you could also compare your approach with ESM-GearNet, and also provide the results of supervised downstream prediction with frozen embeddings in Section 4.2 (as opposed to fine-tuning the model).

- Minor points: $e$ should be replaced by $e_i$ on line 154, and $m$ should be replaced by $M$ in Eq. (6).

**Limitations:**

As the authors allude to, the main limitation of the model is the assumption of the availability of structural data and the computational complexity of the structural quantization process.

---

> ### Author Rebuttal · Authors · 2024-08-07
>
> We sincerely appreciate your thoughtful feedback on our work. We would like to address your questions and concerns as follows:
>
> **Weaknesses**
>
> **W1.** Feeding a "default/noise" structure input alongside the actual sequence was not included in the current version, but we have a solution for it, and this solution is not very sophisticated. We trained ProSST(MST), where MST stands for Masked Structure Training. During pre-training, each sample's structure sequence has a 50% probability of being replaced by a fully masked sequence [1,1,…,1]. This approach simulates the scenario of missing protein structures. Although ProSST(MST) outperformed ProSST(K=0) in sequence-only mode, it was inferior to both the original ProSST model with structure inputs and ProSST(MST) with structure inputs.
>
> We offer two approaches for sequence-only proteins:
>
> - Approach 1: Use AlphaFold or ESMFold to predict structures.
> - Approach 2: Use the ProSST(MST).
>
> We have evaluated these methods on the ProteinGym benchmark and perplexity on the validation set. The results are as follows:
>
> |Model|Structure|ProteinGym|Perplexity|
> |---|---|---|---|
> |ProSST (K=2048)|AlphaFold2|0.504|9.033|
> |ProSST (K=2048)|ESMFold|0.471|9.144|
> |ProSST (MST)|Missing|0.438|10.325|
> |ProSST (MST)|AlphaFold|0.456|9.447|
> |ProSST (K=0)|Missing|0.392|12.190|
>
> Rows 1-2 show the performance of Approach 1. Rows 3-4 show the performance of the new model ProSST(MST). When using AlphaFold, ProSST (MST) is inferior to ProSST (K=2048). We will add a discussion on how to apply ProSST to the revised paper.
>
> **Questions:**
>
> **Q1.** The reason is that **40** covers most cases. We selected the nearest **40** residues within a distance of **10 Å**, according to [1]. As shown in **Figure R1**, local structures with more than **40** neighbors account for only 0.00052% (5.2 x 10^-6), indicating that our choice covers most cases.
>
> **Q2.** They are not the same. $L$ is the number of residues in a protein, also referred to as the length of the protein, while $l$ is the number of nodes in a graph. A protein contains $L$ residues, each corresponding to a local structure, which in turn corresponds to a graph $G$. For any arbitrary graph $G$, $l$ is the number of nodes in it.
>
> **Q3.** The GVP encoder includes a six-layer message-passing graph neural network in which a geometric perceptron replaces the MLP to ensure translational and rotational invariance of the input structure. Our GVP encoder is consistent with the original GVP-GNN [2], except that we removed the residue type information. The GVP encoder is trained from scratch. We will provide detailed descriptions and parameterizations of the GVP in the Appendix.
>
> **Q4.** Both choices are worse than discrete structure tokens. (1) Using the centroid embedding (K=2048) as input yields inferior results:
>
> |Structure Inputs|ProteinGym|DeepLoc|Perplexity|
> |---|---|---|---|
> |Centroid Embeddings|0.462|91.73(±0.24)|9.932|
> |Structure Tokens|0.504|94.32(±0.10)|9.033|
>
> We believe it is because the Transformer requires a learned structure token embedding rather than a fixed embedding.
> (2) Directly using continuous local structure embeddings as structure inputs leads to overfitting, as shown in **Figure R2**. Similar results have also been observed in SaProt.
>
> **Q5.** $l$ represents the total number of nodes in a graph $G$. Average pooling refers to averaging the embeddings of all nodes within the graph. $l$ is not always equal to 40. Because when there are not more than 40 residues within a 10 Å distance, $l$ will be less than 40.
>
> **Q6.** The disentangled attention contains multiple regular attention mechanisms, allowing the model to learn different attention patterns for better contextual representation. The computation of the disentangled attention is also less than that of directly concatenated self-attention because the complexity of the attention mechanism increases quadratically with the sequence length. Although disentangled attention involves additional attention calculations, it does not increase the sequence length.
>
> **Q7.** The $k$ in Eq.(2) should actually be $L_{max}$, which is the cutoff of relative position. We will correct it in the paper.
>
> **Q8.**  (1) Because our model aims to learn the contextual representation of residues rather than structure tokens, structure $s$ in Eq.(6) is left unmasked. (2) When calculating scores of variants, we utilize the structure of wild sequences because mutants only slightly alter the structure of proteins [3], and the structure of mutants is difficult to predict[4]. Other structure models, including ESM-IF, SaProt, ProteinMPNN, etc., also use the wild-type structure when scoring variant sequences.
>
> **Q9.** We have compared ProSST to ESM-GearNet and ProSST(fixed parameters) in fine-tuning downstream tasks. The results are as follows:
>
> |  |  | DeepLoc | Metal Ion Binding | Thermostability | GO-MF | GO-BP | GO-CC |
> | --- | --- | --- | --- | --- | --- | --- | --- |
> | ESM-GearNet | 690M | 93.55 | 74.11 | 0.651 | 0.676 | 0.516 | 0.507 |
> | ProSST(fixed) | 110M | 92.36 (±0.24) | 74.27 (±0.15) | 0.697 (±0.06) | 0.651 (±0.013) | 0.479 (±0.013) | 0.482 (±0.009) |
> | ProSST | 110M | 94.32(±0.10) | 76.37(±0.02) | 0.726(±0.04) | 0.682 (±0.003) | 0.492 (±0.004) | 0.504 (±0.002) |
>
> The scores of ESM-GearNet are derived from SaProt. Note that the slightly different scores compared to the manuscript are due to the updated random seed and dataset. We have re-evaluated ProSST with different seeds to compute the average performance. Additionally, the evaluation was conducted on the updated GO dataset by SaProt.
>
> **Q10**. We will correct them. Thank you.
>
> References
>
> [1] Discovery of Novel Gain-of-Function Mutations Guided by Structure-Based Deep Learning.
>
> [2] Learning from Protein Structure with Geometric Vector Perceptrons.
>
> [3] A folding space odyssey.
>
> [4] Can AlphaFold2 predict the impact of missense mutations on structure?

---

> > ### Comment · Reviewer_t3Vw · 2024-08-10
> >
> > Thank you. I have decided to increase my score in favor of acceptance after reading the rebuttal and the rest of the reviews. I believe the paper presents a worthwhile contribution that is of interest to the NeurIPS audience.

---

> > > ### Author Response · Authors · 2024-08-13
> > >
> > > We sincerely appreciate the time and effort you dedicated to reviewing our paper and reading our responses. Your valuable suggestions and insightful comments have significantly contributed to refining the quality of our work. No matter what the final result will be, the thoughtful communication with you has better clarified our paper's goals and logic. We would like to express our great gratitude to you!

---

### Author Rebuttal · Authors · 2024-08-07

# Rebuttal

We thank all the reviewers for their detailed comments and insightful suggestions. We have incorporated additional experiments and analyses based on the recommendations.

Here, we present a concise overview of the major enhancements that have been universally implemented, focusing on:

- We provide approaches for deploying ProSST on sequence-only proteins, including using AlphaFold or ESMFold to predict structures, or using the new ProSST (MST) model trained on randomly masked structure sequences that support sequence-only input.
- We conducted additional analysis of disentangled attention: (1) The significant decline in ProSST's performance on incorrect and random structure tokens indicates that disentangled attention has learned to leverage structure tokens rather than ignore them. (2) Evidence is provided using ProSST (K=1) to show that the model's improvement is not solely due to disentangled attention. (3) The different types of attentions within disentangled attention are visualized using green fluorescent protein (GFP).
- We performed additional comparisons with more baselines, including adding ProSST (fixed) and ESM-GearNet to supervised fine-tuning downstream tasks, and EVmutation, DeepSequence, WaveNet, RITA, UniRep, ProGen2, and VESPA to zero-shot mutant effect prediction.
- We conducted additional speed analyses, including the training time of the clustering model, the time for structure tokenization, and the inference time of the Transformer model.
- Errors and spelling mistakes have been corrected.
- We reported experimental error for fine-tuning tasks with different seeds, as well as significance tests for zero-shot mutant effect prediction.

The figures are included in the attached PDF, with corresponding references provided in each of our responses. We believe that a brief overview of these additional results will provide clear context for understanding the significance of the updates we have made.

**Figure R1.** The distribution of the number of residues within 10 Å distance.

This figure explains why we choose the nearest 40 residues as the neighborhood of each node. We only build edges for two residues when their distance is less than 10 Å. According to this figure, for an arbitrary residue, the number of residues within 10 Å of it almost never exceeds 40. Therefore, we chose 40 as the threshold, which covers almost all cases.

**Figure R2.** Perplexity curves of pre-training on continuous local structure embeddings.

This figure explains why we do not use continuous local structure embeddings as input: they can cause overfitting.

**Figure R3.** Perplexity curves of ProSST (K=1) and ProSST (K=0).
We trained ProSST (K=1), where the structure tokens are replaced with a constant value of 1. This setup helps preserve the disentangled attention mechanism. Although ProSST (K=1) employs disentangled attention and ProSST (K=0) does not, their training curves show almost no difference. This result indicates that disentangled attention alone cannot enhance the model's performance without correct structure tokens.

**Figure R4.** Different types of attentions on Green Fluorescent Protein (GFP). These attentions are the average of each head in the final layer of the Transformer.

We visualize the attention learned by ProSST on GFP to investigate whether disentangled attention can learn different attention patterns. A significant difference between R2R and S2R can be observed.

**Figure 5.** pLDDT vs. Spearman of ProSST on ProteinGYM.

**Figure 6.** pLDDT vs. Spearman of SaProt on ProteinGYM.

**Figure 7.** pLDDT vs. Spearman of ESM-IF1 on ProteinGYM.

These figures show the relationship between the Spearman's performance in mutant effect prediction and the pLDDT predicted by AlphaFold. There is a positive correlation between pLDDT and model performance.

**Figure 8.** Inference speed of different sequence lengths. (Batch size=16)

We tested the inference speed of ProSST on proteins of different lengths using a batch size of 16 on a server equipped with two Intel 8468C processors and a 3090 GPU.

**Figure 9.** Training and validation curves of the structure encoder.

We present the training and validation loss during the training process of the structure encoder, showing that both losses are stable.

---

### Decision · Program_Chairs · 2024-09-25

**Decision:**

Accept (poster)

**Comment:**

The work introduces ProSST, a novel protein language model that simultaneously leverages sequence and structure information to produce SOTA performance on various widely-used benchmarks. The novelty is in the use of (1) vector-quantized graph autoencoder to tokenize protein structure into local structure tokens and (2) disentagled attention to model the relationship between sequence tokens and structure tokens. The consensus is that the novelty, clarity, and performance introduced in this work will be of interest to NeurIPS audience.